# Management of Primary Female Urethral Adenocarcinoma: Two Rare Case Reports and Literature Review

**DOI:** 10.3390/medicina59010109

**Published:** 2023-01-04

**Authors:** Junjie Tian, Ting Zhu, Zhijie Xu, Xiaoyi Chen, Yunfei Wu, Guanghou Fu, Baiye Jin

**Affiliations:** 1Department of Urology, The First Affiliated Hospital, School of Medicine, Zhejiang University, Hangzhou 310024, China; 2Department of Urology, Jinyun People’s Hospital, Lishui 321400, China

**Keywords:** female urethral adenocarcinoma, mucinous adenocarcinoma, multimodal therapy, follow-up

## Abstract

Primary urethral adenocarcinoma in females is an extremely rare malignancy with unclear origin and only a few retrospective cases have been reported. The controversy continues to exist over the origin of primary urethral adenocarcinoma from periurethral glands (which include the Skene’s glands), urethritis glandularis or intestinal metaplasia. Herein, we report one case of a 49-year-old female with distal urethral adenocarcinoma who presented with obstructive voiding. Abdominal and pelvic CT scans and chest radiology were unremarkable. Biopsy of the mass confirmed urethral adenocarcinoma. The patient underwent partial ureterectomy and was disease-free at the 2-years follow-up period. We also present another extremely rare case of primary urethral adenocarcinoma with mucinous features in a 58-year-old female who initially complained of external urethral orifice itching with painless urethral bleeding and was treated with local excision. The patient has not received any neoadjuvant or adjuvant therapy, and experienced tumor recurrence, inguinal lymph nodes metastasis, and even local iliopsoas metastasis during over 10-years follow-up. In conclusion, our current study emphasizes the importance of imaging studies and biopsy in making an accurate preoperative diagnosis of this rare disease, and further highlights the role of multimodal therapy. A combination of radiotherapy, chemotherapy and surgery is recommended for the optimal local and distant disease control. Moreover, better medical compliance and regular follow-up are required in these patients.

## 1. Introduction

Urethral carcinoma in female is a rare malignancy, accounting for approximately 0.02% of all female cancers and less than 1% of cancers in the female genitourinary tract [1]. Of the female urethral carcinomas, squamous cell carcinoma is the most common histological type, accounting for 70% of all types followed by transitional cell carcinoma (20%) and adenocarcinoma (8–10%) cell types [2]. Primary urethral adenocarcinoma in females is an extremely rare malignancy with unclear origin, which has been categorized into two main histological subtypes: columnar/mucinous and clear cell [3]. Hitherto, controversy continues to exist over the origin of primary urethral adenocarcinoma from periurethral glands (which include the Skene’s glands), urethritis glandularis or intestinal metaplasia [3,4,5,6,7]. Primary urethral adenocarcinoma typically presents with vague symptoms. At initial presentation, the symptoms of an extraurethral mass, bladder outlet obstruction, pelvic pain, urethrocutaneous fistula, abscess formation, or dyspareunia usually lead to discovery of the advanced tumor and a poor outcome [8,9,10]. Physical examination, cystourethroscopy, and CT/MRI imaging of the abdomen and pelvis are usually applied for the initial assessment of females with suspected urethral carcinoma, and biopsy is needed to make an accurate preoperative diagnosis of this rare disease [10]. The prognosis is determined mainly by the clinical stage, pathological grade, status of lymph nodes, presence of distant metastasis, and type and modality of treatment [11].

There were only a few retrospective cases reported about the rare primary adenocarcinoma of the female urethra. Herein, we report two cases of female urethral adenocarcinoma, including an extremely rare case of mucinous adenocarcinoma with inguinal lymph nodes metastasis. Our current study emphasizes the importance of imaging studies and biopsy in making an accurate preoperative diagnosis of this rare disease, and further highlights the role of multimodal therapy; a combination of radiotherapy, chemotherapy and surgery is recommended for the optimal local and distant disease control. Moreover, better medical compliance and regular follow-up are required for these patients. 

## 2. Case Presentation

### 2.1. Case 1

A 49-year-old female patient presented with obstructive voiding over the previous 1 month. Physical examination showed a pedicled mass measuring approximately 0.8 × 1.2 cm in diameter without bleeding or tenderness at the external urethral orifice. No enlarged inguinal lymph nodes were palpated. The results of routine hematology, blood chemistry, and tumor serum markers studies were normal. CT scans of the abdomen and pelvis, and chest radiology showed no abnormality. Biopsy of the mass confirmed urethral adenocarcinoma (Figure 1A). The patient underwent repeat cystoscopy and distal partial urethrectomy. Subsequent routine histopathology of hematoxylin and eosin (H&E) revealed an adenocarcinoma of the urethra orifice with negative margins, which was consistent with the previous biopsy pathology result (Figure 1B). It was determined as T1N0M0 according to the 8th edition of the TNM (Tumor Node Metastasis) classification. The patient did not receive any other adjuvant therapy due to the current localised TNM stage. The patient recovered without complications and maintained normal urination function. During the 2-years follow-up period, the patient remained free of local recurrence and distant metastasis under the local hospital’s review processes.

### 2.2. Case 2

A 58-year-old female patient first presented to our medical center in 2010, who had no significant past medical history and presented with repeated external urethral orifice itching over the previous 1 year and with painless occasional urethral bleeding over the previous 3 months. Vaginal examination showed a pedicled mass measuring approximately 1.0 × 1.5 cm at the external urethral orifice at 11 o’clock position, with ulcerated bleeding on the surface, obvious tenderness, and medium texture. No enlarged inguinal lymph nodes were palpated. The results of serum projects were in the normal range and urinalysis showed a large number of red blood cells per high power field. A puncture biopsy of the mass showed villiform adenocarcinoma (mild to moderate atypia) before admission in a local hospital. The patient underwent urethral orifice tumor resection under local infiltration anesthesia. Subsequent routine pathology confirmed the urethral adenocarcinoma with mucinous features (moderate atypia) (Figure 2). The patient recovered without complications, maintained normal urination function, and did not receive other adjuvant therapy due to the current localised TNM stage (T1N0M0). However, during a postoperative 5-years follow-up time of case 2, the periodic routine follow-up examinations was not properly conducted, which was not enough to determine whether the patient had local recurrence or distant metastasis.

With the development of a tumor, the patient was re-admitted with a change in the urination path and external urethral orifice itching over the previous 1 month in the 6th year after the initial operation (in 2016). Vaginal examination found a pedicled mass measuring approximately 3.0 × 1.5 cm in the external urethral orifice at 11 o’clock position, without tenderness and bleeding. In addition, one enlarged lymph node about 3.0 cm in diameter was palpated in the right inguinal region, which was hard and mobilized but free of tenderness. Meanwhile, cystourethroscopy also showed a pedicled mass measuring approximately 3.0 × 1.6 cm at the external urethral orifice (Figure 3A). An enhanced pelvic computed tomography (CT) scan indicated a 3.1 × 1.9 cm solid cystic mass adhering on the external urethral orifice, and uneven enhancement after contrast administration (Figure 3B). The right groin had an elliptical high-density shadow, about 3.0 × 1.5 cm, with multiple spot-like calcifications seen inside, and none enhancement after contrast administration. Besides, there was an enlarged lymph node shadow beside the right iliac blood vessel, about 1.6 × 0.7 cm (Figure 3C). The results of routine blood tests and further blood chemistry showed no abnormal indicators ([which included the normal range of carcinoembryonic antigen 19-9 (CA19-9), α-fetoprotein (AFP) and prostate-specific antigen (PSA)). The patient was treated with extended urethral mass excision to ensure an adequate margin, and the biopsy of the right enlarged inguinal lymph node was also performed intraoperatively. Subsequent routine pathology of H&E reported the invasive mucinous type of urethral adenocarcinoma (moderate to well differentiation) (Figure 3D,d), which was consistent with the previous pathology report. In addition, the H&E pathology of the right inguinal enlarged lymph node biopsy also confirmed metastatic mucinous adenocarcinoma (Figure 3E,e1). Immunohistochemically, the tumor cells were positive for CK7, CK20, CDX2, MUC1, MUC2, MUC5AC and MUC6, while GATA-3 and p63 were negative (Figure 3e2–e4). Considering the poor prognosis of the patient with surgery alone, we recommended that the patient receive further local radiotherapy or systemic chemotherapy to improve prognosis under the current clinicopathological stage (T3N2M0). However, the patient rejected our proposal and recovered gradually in 1 week and healed satisfactorily.

The patient had not received other adjuvant therapy such as radiotherapy and chemotherapy since surgical treatment in 2016 and also without regular review. During the period, lymph nodes in the right inguinal region were gradually affeccted (enlarged lymph nodes 3.0 × 1.5 cm in diameter in 2018). Unfortunately, the patient was admitted again in 2020 due to the enlarged lymph nodes in the bilateral inguinal regions (approximately 6.0 × 4.0 cm in diameter on the right side and 3.0 × 3.0 cm on the left), accompanied by paroxysmal tingling and numbness in the surrounding skin. Pelvic magnetic resonance imaging (MRI) with contrast revealed enlarged lymph nodes with even enhancement in the bilateral inguinal regions (Figure 4A). In addition, there was an abnormal mass signal shadow with approximate dimensions of 3.4 × 2.0 cm in the right iliopsoas muscle, with equal signal in T1- weighted images and higher in T2- and even enhancement after contrast administration (Figure 4B). Chest radiology revealed multiple high-density nodules in both lungs, which were considered metastatic nodules first (Figure 4C). The results of routine hematology, chemistry blood, and tumor serum markers studies were all in the normal range. 

Therefore, an internal multidisciplinary consensus meeting was further discussed for the patient, with input from urology, medical oncology and radiation oncology, as well as pathology and diagnostic radiology. We strongly recommended that the patient underwent enlarged anterior pelvic exenteration with adjuvant postoperative radiotherapy and chemotherapy to further improve prognosis. The patient hoped to improve the quality of life as the first choice, and we respected all her decisions. After adequate communication and careful preoperative preparation, the patient underwent bilateral inguinal lymphadenectomy and right iliopsoas mass lumpectomy. Postoperative pathology reported metastatic mucinous adenocarcinoma were present in the bilateral inguinal lymph nodes (Figure 4D,d) and the right iliopsoas mass (Figure 4E,e). The final clinicopathological stage was identified as T4N2M1 according to the imaging examination and pathologic results. The patient recovered gradually in 5 days and healed satisfactorily. Regrettably, it was learned that the patient still had not received radiotherapy and chemotherapy or any regular review after discharge during the telephone follow-up at 12 and 24 months. Therefore, we are currently unable to comprehensively assess whether the patient has local recurrence again or more distant metastasis.

## 3. Discussion

Primary female urethral carcinoma is a quite rare malignancy which accounts for under 0.003% of all urogenital tract malignancies in females [12]. A large epidemiologic study from the Surveillance Epidemiology and End Results (SEER) database reported that the highest incidence of primary urethral carcinoma was in the 75-year-old age group (7.6 per million). The incidence based on age standardization was 4.3 per million in males and 1.5 per million in females [13]. 

Of the female urethral carcinomas, the most common histological type is squamous cell carcinoma which accounts for 70% of all types followed by transitional cell carcinoma (20%) and adenocarcinoma (8–10%) subtypes. Besides, neuroendocrine carcinoma, paragangliomas, metastasis, sarcomas, lymphoma and melanoma are the other rarer histologic types [2]. 

It was reported that recurrent urinary tract infections [14,15,16] and urethral diverticula [17] were the main etiological factors of female urethral carcinoma. Furthermore, other etiology and risk factors including leukoplakia, polyps, caruncles, childbirth and human papillomavirus infections or other virus infections might also make patients prone to urethral tumor changes.

Primary adenocarcinoma of the urethra in females is an quite rare malignancy of unclear origin, which has been categorized into two main histological subtypes: columnar/mucinous (“intestinal”) and clear cell [3]. The appearance of the histology in mucinous-type adenocarcinoma has a similarity to mucinous carcinoma of the colon and rectum, which is mainly made up of colonic glandular epithelium and might include a mucinous component characterized by the abundant extracellular mucin that is exhibited in parts of the area [18], as in case 2. 

The origin of primary urethral adenocarcinoma remains controversial. One origin of primary urethral adenocarcinoma in females is thought to be the periurethral Skene’s glands, which are regarded as being functionally homologous to the prostate [6,19]. Although a positive PSA test result is considered as evidence of the origin from Skene’s glands [20], other studies have reported several PSA-negative adenocarcinomas that favored an origin from Skene’s glands, where cytochemical and immunohistochemical studies have found similarity to normal Skene’s glands [21,22]. Other studies have reported that chronic irritation of the urethral mucosa can cause intestinal metaplasia, or glandular urethritis, and thus alternative theories of origin have been proposed [5,23].

Most patients with urethral carcinoma presented with obstructive voiding, urinary frequency, hematuria, and palpable extraurethral mass or induration, which account for over 50% of presenting symptoms. Therefore, urethral tumors should be cautiously suspected in any healthy female in the middle age group with urinary retention but with no prior history of urinary tract problems. A small lesion through prolapse of the urethral orifice or a small submucosal bulge on the anterior wall of the vagina can also be seen at initial presentation among some patients. Tumors usually spread by local extension and may suffer ulcers when the tumor develops into the adjacent skin and vulvar area. Furthermore, proximal urethral lesions may progress into the bladder or invade the adjacent vaginal region posteriorly. Lymphatic spread is unusual at the seedling stage, but one-third of patients have clinically palpable nodes at initial state and half with advanced and proximal urethral carcinomas, as in current case 2. Hematogenous extension may occur sequentially to the lung, liver, bone and brain in turn [24]. 

The comprehensive assessment of females who are suspected with urethral carcinoma mainly includes physical checkup under anesthesia, urethrocystoscopy, ultrasonography, chest radiologic imaging, pelvis and abdomen as well as serum studies. Radiologic imaging of urethral carcinoma not only evaluates the extent of local tumor, but also aids the determination of lymphatic and distant metastatic spreading. More and more evidence indicates that CT is inferior to MRI in tumor identification and staging accuracy. It is reported that local urethral tumors in 90% of patients can be accurately assessed via MRI. CT of the abdomen and thorax is also necessary in patients who are suffering invasive disease [25]. In addition, a recent study reported that high-sensitivity PET combined with the increasing tissue resolution of MR (PET/MR) might improve the detection of pelvic and abdominal lesions [26]. Transvaginal ultrasonography can also provide clues to the diagnosis of urethral carcinoma [27]. Biopsy of a suspected tumor is key to a definitive diagnosis. It is essential to exclude metastatic lesions emanating from the presence of colorectal or gynaecologic malignancies for the diagnosis of primary urethral adenocarcinoma.

Due to the rarity and disease heterogeneity of urethral carcinoma, there are limited systematic studies on long-term prognostic follow-up. As for the primary urethral carcinoma, its long-term survival after surgery is mainly referenced from the RARECARE (The project Surveillance of Rare Cancers in Europe) project [28] and the large SEER database [13] and results often cover early period. Certain recent studies also have reported differences in survival outcomes among different clinical stages and treatment modalities [29,30,31,32]. The prognostic outcome is determined largely by the clinical stage and pathological grade, distant metastasis presence and nodal stage, as well as type and modality of treatment [33]. Distal urethra tumors are apt to have a better clinical outcome [8]. In the current case 1, a tumor located at the urethral orifice underwent distal partial urethrectomy, determined as T1N0M0 via the TNM staging system, with regular postoperative follow-up, and had a good prognosis. However, the diagnosis and treatment of case 2 was less satisfactory. The patient had a good prognosis after undergoing urethral tumor resection following initial disease discovery (T1N0M0), but was not regularly followed up after the first operation; after the tumor recurred with local inguinal lymph node metastasis (T3N2M0), the patient underwent local surgery again but refused the multimodal adjuvant therapies. As the tumor progressed and worsened, palliative care was the last option (T4N2M1). It was regrettable that the lack of reasonable medical compliance and reliable monitoring follow-up created the poor prognosis for case 2.

Treatments of female urethral carcinoma are manifold, mainly including surgery, radiotherapy and chemotherapy, either in combination or alone. In recent years, treatment has the tendency towards multimodal options and drawing on multidisciplinary discussions [34]. For localized urethral carcinoma in females, the primary treatment of urethra-sparing surgery (such as transurethral local resection or laser) [35,36] or radiotherapy [37] has been evidenced to give results with satisfactory function, especially for the primary anterior urethral lesions, as in the first case described in our current study. For advanced urethral carcinoma, multimodal treatment combined with radiotherapy, chemotherapy and surgery have been defined as optimal choice for local and distant disease control. Among them, a cisplatin-based polychemotherapeutic regimen has been recommended first for its optimistic effect in advanced primary urethral carcinoma [38], improving survival even in lymph nodes–positive patients [39]. In locally advanced squamous cell carcinoma of the urethra, the combination of curative radiotherapy with radiosensitising chemotherapy prior to surgery is an effective option for improving survival [40,41,42]. However, our second case patient brought a challenging management scenario. The lack of reasonable medical compliance and reliable monitoring follow-up created a management dilemma in the absence of multimodal therapies, which led to regrettable disease progression and exacerbation.

There is no doubt that the presence of positive surgical margins (PSMs) will lead to worse clinical outcomes for cancer patients, bringing challenges in decision-making regarding both diagnosis and treatment. Notably, researchers evaluated the impact of each PSMs site on clinical outcomes after radical cystectomy in bladder cancer patients; urethral PSMs were an ominous sign at the advanced pathological stage and gave a worse prognosis [43,44,45]. Distal urethra tumors are apt to have a better clinical outcome. Transurethral resection or laser is used for small distal urethral tumors, but has also resulted in considerable local failure rates of 16%, with a cancer-specific survival rate of 50% [35]. This emphasized the critical role of local tumor control with negative surgical margins in patients with distal urethral tumor to prevent local and systemic progression [35]. Proximal female urethral carcinomas are more likely to be high stage. Patients with positive proximal margins had a higher risk of progression. The proximal female urethral carcinomas may extend into the bladder or invade the adjacent vaginal region posteriorly. Therefore, it is necessary to emphasize the prognostic value of negative surgical margins at the time of urethral tumor resection.

Along with histological staging, the application of immunohistochemistry in clinical practice is growing. Several molecular markers predict clinical outcomes of the curative treatment and can identify patients who would benefit from additional treatment (conventional or targeted therapies/immunotherapy) [46,47]. Therefore, optimizing patient selection through novel biomarkers and advanced imaging might improve tumor stratification and drive treatment towards personalised strategies [48]. The results of pathological biopsy in the two patients discussed here were urethral adenocarcinoma; the positive results of immunohistochemistry were CK7 (+), CK20 (+), CDX2 (+), MUC1 (+), MUC2 (+), MUC5AC (+) and MUC6 (+), while GATA-3 and p63 were negative in case 2. It was found that CK20 immune response activity was most sensitive to the diagnosis of in situ carcinoma [49]. MUC1, MUC2, MUC5AC and MUC6 and other members of the mucins family, have been indicated as havingclosed association between aberrant mucin expression and aggressive behaviors of malignancies [50,51]. P63 could help in the identification of bladder tumors with squamous differentiation, and a high percentage of p63 immunoexpression showed a significant association with low-grade urothelial carcinoma [52]. For a specific marker, current studies have found that NANOG and GATA3 are sensitive markers for urothelial carcinoma, and might be a potential biomarker for early diagnosis of urothelial carcinoma [53,54]. Emerging biomarkers have shown promise but need further validations before entering into clinical practice. There is no evidence of the application of immunohistochemestry-based markers in urethral carcinoma including urethral adenocarcinoma. It is believed that, in the near future, more convincing conclusions will be presented, and cancer-specific markers for molecular diagnostic and therapeutic strategies will be found.

There is no specific guideline for follow-up cases of primary urethral carcinoma including urethral adenocarcinoma. In the absence of high-level data, it seems reasonable to tailor surveillance regimens according to patients’ individual risk factors [10,55]. Therefore, we recommend applying the general postoperative follow-up plan for our current patients consisting of a physical exam, urethrocystoscopy for patients undergoing limited excision, laboratory exams, urinary cytology and cross-sectional imaging performed every 3 to 4 months for the first year and at longer intervals thereafter. 

Due to the extremely low incidence of female urethral adenocarcinoma, there is a lack of standard evidence and experience in the clinical management of this tumor. We acknowledge the limitations of our current study, namely, retrospective analysis and case reports with a low level of evidence. More systematic research is needed to evaluate the pathological features and appropriate clinical management of female urethral adenocarcinoma. 

## 4. Conclusions

The clinical significance, prognosis and optimal treatment of primary female urethral adenocarcinoma are still largely unclear due to its rarity. The function of imaging studies and biopsy in preoperative accurate diagnosis is highly valued and the role of careful clinical examination has been greatly stressed in this study. The optimal multimodal treatment modalities are recommended for patients with different clinical stages. Furthermore, these patients require regular follow-up and tailored surveillance regimens based on the individual risk factors.

## Figures and Tables

**Figure 1 medicina-59-00109-f001:**
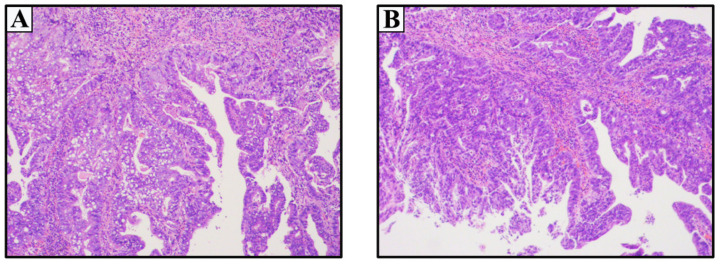
Primary female urethral adenocarcinoma of the Case 1 (4 × 10 magnification). (**A**) Post-biopsy pathology. (**B**) Postoperative routine pathology.

**Figure 2 medicina-59-00109-f002:**
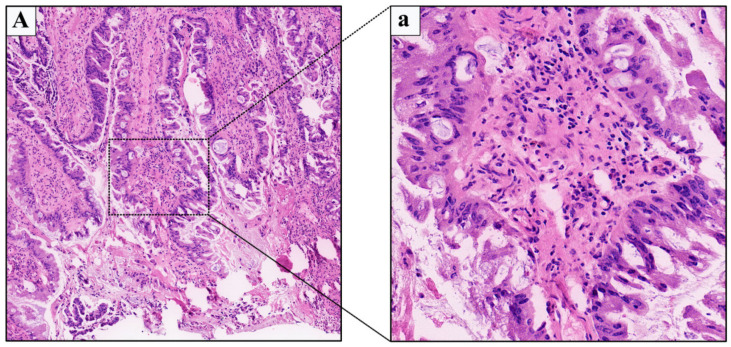
Primary female urethral adenocarcinoma of the Case 2 in 2010. Low-power ((**A**), 4 × 10 magnification) and high-power ((**a**), 40 × 10 magnification) view showing adenocarcinoma of the urethra with mucinous features.

**Figure 3 medicina-59-00109-f003:**
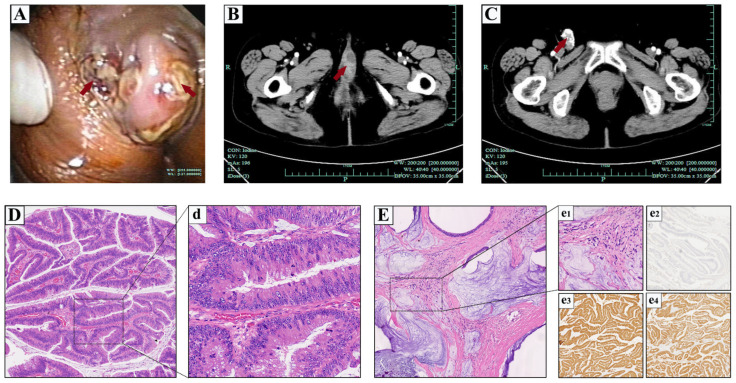
Cystourethroscopic, imaging examinations and histopathologic features of the Case 2 in 2016. (**A**) Cystourethroscopy show a pedicled mass at the external urethral orifice with approximate dimensions of 3.0 × 1.6 cm. (**B**) An enhanced pelvic CT scan indicated a 3.1 × 1.9 cm solid cystic mass adhering on the external urethral orifice, and uneven enhancement after contrast administration. (**C**) Enlarged lymph node with even enhancement were observed beside the right iliac blood vessel. Pelvic MRI demonstrated lesion surrounding the urethra. (Red arrows indicate the corresponding foci). H&E staining of low-power ((**D**), 4 × 10 magnification) and high-power ((**d**), 40 × 10 magnification) view showing the invasive mucinous type of primary urethral adenocarcinoma. Hematoxylin and eosin (H&E) staining of low-power ((**E**), 4 × 10 magnification) and high-power ((**e1**), 40 × 10 magnification) view showing the right inguinal lymph node biopsy pathology confirming metastatic mucinous adenocarcinoma. Low-power view showing GATA-3 immunostain ((**e2**), 4 × 10 magnification, negative), CK20 immunostain ((**e3**), 4 × 10) magnification, strong positive), and CDX2 immunostain ((**e4**), 4 × 10 magnification, strong positive).

**Figure 4 medicina-59-00109-f004:**
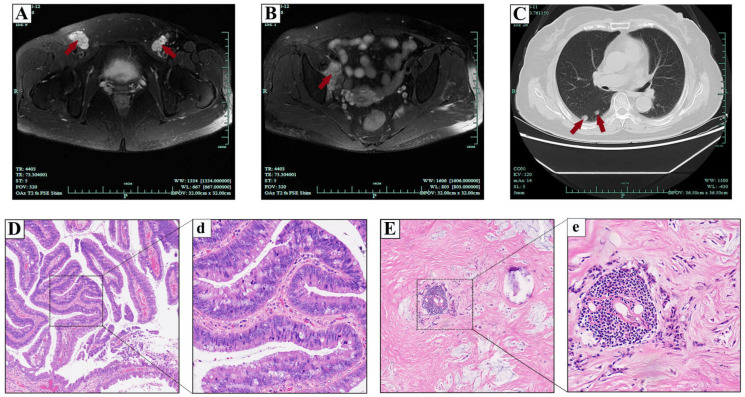
Imaging examinations and histopathologic features of the Case 2 in 2020. (**A**) Enlarged lymph nodes with marked enhancement were seen in the bilateral inguinal regions. (**B**) An abnormal mass signal shadow with approximate dimensions of 3.4 × 2.0 cm in the right iliopsoas muscle, with equal signal in T1- weighted images and higher in T2- and even enhancement after contrast administration. (**C**) Pulmonary CT scan indicated multiple high-density nodules in both lungs, which were considered metastatic nodules first. (Red arrows indicate the corresponding foci). H&E staining of low-power (4 × 10 magnification) and high-power (40 × 10 magnification) view showing the metastatic mucinous adenocarcinoma were present in the bilateral inguinal lymph nodes (**D**,**d**) and right iliopsoas mass (**E**,**e**).

## Data Availability

Not applicable.

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
