# Peer review of "Management of Primary Female Urethral Adenocarcinoma: Two Rare Case Reports and Literature Review"

_medicina, 2023, doi:10.3390/medicina59010109_

Round 1
Reviewer 1 Report
The case report is interesting, and the topic is rare. Abstract and intro are well written.
- I would move the ethics after the introduction.
- Please give better details on pathology reports: TNM, immunochemistry, etc
- Please explain the patient follow-up and treatment after urethrectomy in case 1 and 2 (presentation in 2010)
- I would not divide the case two in “presentation in 2010 and 2016, 2020 to date” since it’s the same case. Please describe it all in the same subchapter “case 2”
- Please insert the limitations of the case reports at the end of the discussion
Reviewer 2 Report
In this manuscript the authors presented an interesting case series about primary adenocarcinoma of the urethra in female pts. The topic is quite remarkable as the literature is quite limited to retropsective case series. Limited data are available considering the treatment strategy. No neoadjuvant treatment were adopted. Thus, no alteration of molecular markers were imposed by neoadjuvant admistration. Moreover, no adjuvant treatments were administrated. Follow-up presented was large. The information provided are relevant to the topic.
Some comments below.
What guidelines were followed considering the follow-up setting? Were the pts included in this report further discussed within an internal multidisciplinary consensus meeting?
The authors highlighted the importance of surgical margins status at time of resection. Please considering to further discuss this relevant topic considering the following literarature (doi: 10.1007/s00345-021-03776-5; doi: 10.3390/cancers14235740).
The authors concluded that a multimodal treatment modalities are recommended for patients with such a disease within different clinical stages. The authors should consider to discuss the impact of immunohistochemestry - based markers as a further tool to guide the treatment strategy. Conventional and novel IHC-based biomarkers have been found to improve risk stratification among different malignancies across different setting: radical treatment only, neoadjuvant and adjuvant settings including conventional agents and novel targeted therapies (doi: 10.1016/j.urolonc.2021.10.010; doi: 10.1016/j.euo.2021.04.004; doi: 10.1016/j.ajur.2021.05.001). These papers are worth mentioning in order to improve the scientific sound and the multidisciplinary content together with the future perspectives carried by this manuscript.
Conclusions: I would add a statement about the importance of multidisciplinary consensus meeting for the management of these patients.
Round 2
Reviewer 2 Report
The authors revised the manuscript properly. Congratulations.